# Ionomic Combined with Transcriptomic and Metabolomic Analyses to Explore the Mechanism Underlying the Effect of Melatonin in Relieving Nutrient Stress in Apple

**DOI:** 10.3390/ijms23179855

**Published:** 2022-08-30

**Authors:** Yang Cao, Peihua Du, Jiahao Ji, Xiaolong He, Jiran Zhang, Yuwei Shang, Huaite Liu, Jizhong Xu, Bowen Liang

**Affiliations:** College of Horticulture, Hebei Agricultural University, Baoding 071001, China

**Keywords:** apple, melatonin, nutrient stress, ionomic, transcriptomic, metabolomic

## Abstract

Nutrient stress harms plant growth and yield. Melatonin is a biologically active, multifunctional hormone that relieves abiotic stress in plants. Although previous studies have shown that melatonin plays an important role in improving nutrient-use efficiency, the mechanism of its regulation of nutrient stress remains unclear. In this study, melatonin was applied to apple plants under nutrient stress, and morphological indices, physiological and biochemical indices, and stomatal morphology were evaluated. The response of apple plants to nutrient deficiency and the melatonin mechanism to alleviate nutrient stress were analyzed by combining ionome, transcriptome, and metabolome. The results showed that exogenous melatonin significantly alleviated the inhibitory effect of nutritional stress on the growth of apple plants by regulating stomatal morphology, improving antioxidant enzyme activity, promoting ion absorption, and utilizing and changing the absorption and distribution of minerals throughout the plant. The transcriptome results showed that melatonin alleviated nutrient stress and promoted nutrient absorption and utilization by regulating glutathione metabolism and upregulating some metal ion transport genes. The metabolome results indicated that levels of oxalic acid, L-ascorbic acid, anthocyanins (cyanidin-3-O-galactoside), lignans (lirioresinol A and syringaresinol), and melatonin significantly increased after exogenous melatonin was applied to plants under nutrient stress. These differentially expressed genes and the increase in beneficial metabolites may explain how melatonin alleviates nutrient stress in plants.

## 1. Introduction

High-quality arable land is limited and mostly used to produce food crops. China’s apple-producing areas are primarily located in mountainous and hilly areas with poor soils. Coupled with nutrient loss caused by impractical farming systems, soil nutrient imbalances and shortages in orchards are widespread. Adequate and balanced mineral nutrition is an important prerequisite for ensuring good growth and development of plants [1]. Macroelements (N, P, K, S, Ca, and Mg) and microelements (Fe, Mn, B, Zn, and Cu) play indispensable roles in plant growth and development. For example, a lack of N leads to a decrease in plant chlorophyll content, inhibits photosynthesis, decreases dry-matter accumulation, and alters the root configuration [2,3,4]. P deficiency reduces photosynthetic capacity, darkens leaves, causes leaf veins to become purplish-red, increases the root-to-crown ratio, and augments the secretion of organic acids and phenols in roots [5]. When plants are deficient in K, older leaves and leaf margins often turn yellow, and brown spots or patches appear. It also inhibits photosynthesis and reduces resistance to stress [6]. The transcriptome and metabolome have been combined to analyze the regulatory response mechanism of apples to different N conditions. The results confirmed that apple plants adapt to different N environments by regulating carbon metabolism, N metabolism, and the flavonoid pathway [7]. More attention should be paid to the observation that a nutrient deficiency in an apple orchard is often not a single but rather a complex problem in which multiple elements may be lacking at the same time [8], which may lead to complex stress responses with elusive regulatory mechanisms. Improving management methods and replenishing fertilizer is the main way to solve nutrient deficiencies, but this undoubtedly increases the cost of investment. Therefore, new green and effective ways to improve plant utilization efficiency are needed under limited nutrient conditions.

Melatonin is a pleiotropic molecule in plants that is synthesized by chloroplasts and mitochondria and exists in almost all plant tissues, where it plays an important role in the regulation of stress resistance [9,10]. Melatonin not only protects plants by scavenging reactive oxygen species (ROS) in vivo but also has other functions [11]. Previous studies have reported that melatonin can improve the formation of apple adventitious roots, accelerate the invisible propagation of apple rootstocks that are difficult to root [12], and improve the quality of apple fruit [13]. In addition, it protects the photosynthetic system and enhances photosynthetic efficiency, regulates the activity of antioxidant enzymes [14,15], and coordinates hormone levels in plants to enhance their resilience [16,17,18]. Recent studies have reported that melatonin promotes the absorption and utilization of minerals in plants, thus alleviating the inhibitory effect of abiotic stress on plant yield [6,19]. In a previous study, we revealed that melatonin significantly alleviates the inhibitory effect of NO_3_^−^ and NH_4_^+^ deficiencies by regulating the activity of N metabolic enzymes and expressing N transport and metabolic genes [3,4]. However, the mechanism of how melatonin regulates nutrient stress response is relatively unknown.

Transcriptomics and metabolomics are high-throughput omics methods that efficiently respond to the regulation of gene expression and metabolite levels within plant tissues or cells and have been widely used by researchers to study plant responses to different stressors [7]. The perception of abiotic stress, such as nutrient stress, further leads to the expression of related genes, including those related to metabolic pathways, such as drought, that can affect the expression of genes related to the abscisic-acid-synthesis pathway. In addition, the metabolite changes are a direct reflection of the physiological state of a plant. For example, oat roots metabolize citric acid and malic acid to adapt to phosphorus deficiency [20]. As minerals are involved in many key metabolic pathways, a lack of minerals has numerous effects on metabolism. For example, K is primarily involved in osmotic regulation and maintenance of the cation–anion balance [21]. Ca is a ubiquitous and versatile second messenger that is involved in many metabolic pathways [22,23]. Mg is essential for chlorophyll, and P is a key component of ATP, nucleic acids, and phospholipids [24]. Transcriptome and metabolome studies on the effects of melatonin on apple leaf genes and their metabolites under nutritional stress will help to explain the mechanism of the response to nutrient stress and how melatonin improves absorption and utilization of nutrients and alleviates nutritional deficiencies in apples.

Applying some substances can enhance the ability of plants to resist abiotic stress, which is a complex process involving genes, signaling pathways, and changes in metabolites. However, few studies have conducted transcriptomic and metabolomic analyses on the alleviation of nutrient stress in apples with exogenous melatonin. In this study, physiology, ionome, transcriptome, and metabolome analyses were used to study the response of apple plants to nutrient deficiency and to explore the mechanism of melatonin-induced tolerance and adaptability of apple seedlings to nutrient deficiency. The results will provide a theoretical basis for reducing fertilizer use and improving the nutrient utilization efficiency of apples.

## 2. Results

### 2.1. Exogenous Melatonin Regulates the Growth and Development of Apple Plants and Activates the Antioxidant System under Nutrient Stress

Plant growth was severely inhibited under nutrient stress, and exogenous melatonin significantly alleviated the inhibitory effect of nutrient stress on growth (Figure 1a). After 20 days of hydroponic treatment, the plant lengths (PLs), trunk diameters (TDs), total dry weights (TDWs), and relative growth rates (RGRs) of the ST (1/20 Hoagland nutrient solution treatment) plants had decreased by 20.3%, 20.3%, 24.6%, and 49.7%, respectively, compared to CK (1/2 Hoagland nutrient solution treatment). However, exogenous melatonin significantly alleviated the downward trend in these indicators. The indicators were increased by 25.0%, 21.1%, 33.9%, and 97.1% compared to ST, respectively. In addition, the root/stem ratios (RSRs) of the MST (1/20 Hoagland nutrient solution with 0.1 μmol/L melatonin treatment) plants were significantly higher than those of the ST plants (Figure 1b).

In addition, nutrient stress significantly inhibited root growth (Figure 2a). Root lengths, diameters, surface areas, volume, tips, and forks of ST plants decreased by 23.9%, 18.5%, 31.2%, 47.3%, 33.6%, and 33.0%, respectively, compared to CK. Exogenous melatonin significantly increased the roots and architecture indices, by 26.6%, 15.9%, 43.7%, 57.6%, 37.5% and 43.7%, respectively, compared to ST (Figure 2b).

At the same time, total chlorophyll content (TCC) in ST plants decreased by 25.7% compared to CK after 20 days of hydroponic treatment but only decreased by 5.7% after exogenous melatonin application (Figure 3a). Malondialdehyde (MDA) content, relative electrical conductivity (REL) of leaves, and root activity were determined. MDA and REL in ST plant increased by 17.5% and 13.8% compared to CK, respectively. Root activity (TTC) decreased by 15.2% in ST plants, while MDA and REL increased by only 9.0% and 1.7%, and root activity decreased by 6.3% in MST plants (Figure 3b–d). Exogenous melatonin activated the antioxidant systems and improved the antioxidant capacities of plants under nutrient stress. Superoxide dismutase (SOD), peroxidase (POD), catalase (CAT), and ascorbate peroxidase (APX) activity in MST plants increased by 13.4%, 16.8%, 49.0%, and 26.3%, compared to ST, respectively (Figure 3e–h).

### 2.2. Exogenous Melatonin Mediates the Regulation of Stomatal Configuration under Nutrient Stress

After 20 days of the hydroponic nutrient-deficiency treatment, the lower surfaces of the leaf samples were scanned at 700× and 2500× magnifications, respectively (Figure 4a). Nutrient stress and exogenous melatonin changed the stomatal structure. Although exogenous melatonin had no significant effect on stomatal width, stomatal density decreased by 17.4%, and stomatal length and stomatal opening increased by 13.9% and 116.7%, compared to ST plants, respectively (Figure 4b).

### 2.3. Exogenous Melatonin Mediates the Regulation of the Concentration, Uptake, Transfer, and Distribution of Minerals in Plants under Nutrient Stress

When plants were stressed by nutrient deficiency, the concentrations of almost all macro- and microelements significantly decreased. The concentrations of N, P, K, Ca, Fe, Mn, Cu, Zn, and B significantly decreased in ST plants by 13.1%, 14,2%, 15%, 16.1%, 19.2%, 26.1%, 36.6%, 17.8%, and 16.6% compared to CK, respectively. Exogenous melatonin significantly alleviated the decreases in N, P, K, Fe, Mn, Zn, and B by 1.1%, 6.7%, 8.2%, 4.8%, 4.7%, 3.4%, and 8.6%, respectively (Appendix A).

The absorptions of N, P, K, Ca, Mg, Fe, Mn, Cu, Zn, and B were significantly reduced by 58.8%, 59.0%, 58.1%, 55.1%, 51.9%, 65.3%, 73.5%, 71.6%, 57.1%, and 60.4%, respectively, in ST plants compared to CK. However, exogenous melatonin significantly increased the absorptions of these elements. The uptake of these minerals in MST plants increased by 131.4%, 115.9%, 117.1%, 103.2%, 116.0%, 141.0%, 224.2%, 100.0%, 111.8%, and 122.2%, compared to ST, respectively (Figure 5).

The transport rates of macro- and microelements were severely affected by the nutrient stress. The nutrient transport rates of N, P, K, Ca, Fe, Mn, and B to leaves were significantly higher in MST than in ST plants. In addition, the root accumulation rates of K, Mg, Fe, Mn, Zn, and Cu were significantly higher in MST plants than in ST plants, but no significant differences in the accumulation rates of N, P, Ca, or B were observed in roots between the two groups. The root accumulation rates of the minerals were consistent with the mineral transport rates to leaves (Figure 6).

The concentrations of N, P, K, Ca, Mg, Fe, Mn, Zn, Cu, and B were highest in leaves, and applying 0.1 μM melatonin significantly increased the partitioning of almost all minerals in the leaves and roots. N, P, K, Mg, Fe, Mn, and B levels increased in the leaves of MST plants by 19.5%, 7.5%, 19.0%, 14.7%, 25.9%, 129.5%, and 12.0% compared to ST. N, P, K, Mg, Fe, Zn, and Cu levels increased by 19.1%, 13.4%, 14.8%, 20.7%, 23.3%, 35.4%, and 15.3%, respectively, in the roots of MST plants. Compared to ST, N, P, K, Fe, Zn, Cu, and B levels increased in the stems of MST plants by 16.3%, 25.1%, 13.1%, 17.9%, 104.2%, 12.1%, and 32.1% (Figure 7).

### 2.4. Differentially Expressed Gene Analysis

Three transcriptomic comparisons (including CK, ST, and MST) were performed to identify the effect of exogenous melatonin on differentially expressed genes (DEGs) in apple leaves under nutritional stress. Three pairs of libraries (ST/CK, MST/CK, and MST/ST) were prepared according to the different treatments, and transcriptome analysis was performed on apple leaves (three replicates per treatment). A total of 65.88 Gb of clean data was obtained. Clean data of each sample was ≥ 6 Gb. The percentage of the Q30 bases was ≥94%, indicating that the quality of the transcriptome data was relatively high (Appendix A). PCA was used to detect the severity of changes caused by the different treatments. The results showed an obvious difference among CK, ST, and MST samples (Appendix A). A clustering heat map was used to cluster genes with the same or similar expression patterns. The map indicated that the expression patterns were significantly different between the ST/CK and MST/ST treatments (Figure 8b,d). DESeq2 was used to complete the DEG analysis. The total number of DEGs, the number of upregulated genes, and the number of downregulated genes were counted in each group. A total of 285 (140 up and 145 downregulated) DEGs in MST/CK, 111 (47 up and 64 downregulated) DEGs in ST/CK, and 138 (68 up and 70 downregulated) DEGs in MST/ST were detected (Figure 8a,c and Appendix A). Genes related to chlorophyll synthesis, stress response, metal-ion transport, and phosphate balance changed at the same time in both treatments (Appendix A). These results show that nutrient stress and exogenous melatonin induced transcriptomic changes in apple leaves.

GO and KEGG enrichment analyses were performed to verify the biological functions of the DEGs. The GO analysis exhibited the top 50 enrichment classifications and indicated that most of the genes in ST/CK were involved in biological processes and molecular functions. However, the DEGs in MST/ST were only involved in biological processes (Appendix A). The KEGG enrichment analysis results showed that the DEGs in ST/CK were involved in porphyrin and chlorophyll metabolism, flavonoid biosynthesis pathways, circadian rhythm-plants, and ubiquinone and other terpenoid-quinone biosynthesis pathways. The DEGs in MST/ST were involved in glutathione metabolism and cutin, suberin, and wax biosynthesis (Figure 9).

### 2.5. Validation of DEGs by qRT-PCR

To confirm the RNA-seq results, eleven DEGs that had different roles in plant leaves were selected. The qRT-PCR and RNA-Seq results were consistent for all the ten validated genes (Appendix A), indicating that reliable RNA-seq data were obtained from the samples.

### 2.6. Metabolic Response of Apple Trees to Nutrient Stress

Metabolomic analysis was performed to identify the effect of applying exogenous melatonin to apple plants under nutritional stress. A total of 849 metabolites were detected based on the UPLC-MS platform and a self-built database and were separated into three treatments according to the PCA (Appendix A). We carried out further analysis of the relationships between the different treatments. According to the screening criteria, the numbers of increased and decreased metabolites in MST/CK, ST/CK, and MST/ST were nine and five, two and nine, and nine and seven, respectively (Appendix A). A Venn diagram was prepared to analyze the intersection and unique metabolites between ST/CK and MST/ST. Among them, three different metabolites were common in the ST/CK and MST/CK groups. In all, eight and thirteen different metabolites were unique between ST/CK and MST/ST, respectively (Appendix A). Oxalic acid, 3-ureidopropionic acid, L-asparagine, L-aspartic acid, muconic acid, 2,2-dimethylsuccinic acid, 2′-deoxyinosine-5′-monophosphate, isorhamnetin-3-O-gallate, and solatriose were significantly decreased in ST plants compared to CK (Appendix A). However, oxalic acid, L-ascorbic acid, melatonin, 2-acetyl-3-hydroxyphenyl-1-O-glucoside, 3-O-methylquercetin, cocamidopropyl betaine, lirioresinol A, syringaresinol, cyanidin-3-O-galactoside, and solatriose were significantly increased in MST plants compared to ST (Appendix A). These metabolites were classified into KEGG pathways, and ST/CK was enriched in cyanoamino acid metabolism, beta-alanine metabolism, pantothenate and CoA biosynthesis, and alanine, aspartate, and glutamate metabolism. Moreover, the KEGG pathways enriched in MST/ST included taurine and hypotaurine metabolism and sulfur metabolism (Figure 10).

## 3. Discussion

In this study, nutrient stress resulted in a significant decrease in all growth parameters, including PL, TD, TDW, RSR, and RGR. However, applying exogenous melatonin significantly alleviated the inhibitory effect of nutrient stress on plant growth (Figure 1).

Leaf yellowing caused by reduced chlorophyll content is a symptom of nutrient stress, which drastically reduces photosynthesis (Figure 3a). In our study, applying exogenous melatonin enhanced the absorption and utilization of N, P, and Fe, thus maintaining chlorophyll content at a high level (Figure 5). Nutrient stress significantly reduced stomatal length, width, and aperture (Figure 4), which may lead to lower CO2 assimilation and transpiration rates and a lower photosynthetic rate. In addition, it has been well documented that K-deficient plants have inhibited stomatal movement, enzyme activities, and protein synthesis, while producing large quantities of ROS that impair the photosynthesis process [21]. Here, exogenous melatonin significantly improved stomatal status and the absorption and utilization of K under nutrient stress, which was conducive to maintaining strong transpiration and photosynthetic rates. Chloroplasts are the primary site of melatonin synthesis and one of the organelles most susceptible to ROS [25]. Many studies have reported the protective effects of melatonin on chloroplast and stomatal structure [18,26]. In this study, adding exogenous melatonin significantly reduced the REL and MDA of plant leaves, and increased SOD, POD, CAT, and APX activities (Figure 3). These results indicate that exogenous melatonin reduced the degree of membrane lipid peroxidation in leaves and maintained membrane permeability under nutrient deficiency. Applying melatonin to rice seedlings enhances their tolerance to cold by improving the efficiency of the photosystems and the activity of antioxidant enzymes [27]; our results are consistent with this concept. The strong antioxidant function of melatonin played an important role in the resistance of apple plants to nutrient stress.

Minerals at concentrations of more than 0.1% of plant dry weight are essential for plant growth and nutritional quality [28,29]. Severe mineral deficiencies are usually related to corresponding phenotypic symptoms in plants [24]. Roots play an important role in the absorption and utilization of nutrients. Root growth is inhibited, and root activity and biomass significantly decrease under nutrient stress [8]. In this study, nutrient stress significantly reduced RSR and root activity, as well as the root length, diameter, surface area, volume, tips, and forks. These indices significantly improved in plants that received exogenous melatonin (Figure 2). Root morphology is extremely sensitive to reduced N and P contents and changes accordingly [2,30]. Apple plants increase K uptake by changing their root morphology [6]. Root growth is stunted when Ca, Mg, Fe, and Mn are deficient [8]. Interestingly, applying exogenous melatonin promoted root growth and elongation, increased the root absorption area, and improved root vitality. Such changes in root morphology contribute to adaptation to nutrient stress [4]. In addition, melatonin enhances nutrient transport and accumulation, which in turn affects the distribution of nutrients throughout the plant [6]. In this study, the concentrations of most minerals significantly decreased in plants under nutrient stress, and adding melatonin maintained a higher concentration of these minerals and changed the absorption and distribution of minerals throughout the plant (Appendix A and Figure 7). At the same time, the absorption and accumulation of these minerals significantly decreased in response to stress, and the decrease was alleviated by applying melatonin (Figure 5 and Figure 6).

Transcriptome sequencing was performed to further explain the gene differences caused by nutrient stress and melatonin. Particular numbers of DEGs were detected in ST/CK and MST/ST (Figure 8). According to the KEGG-pathway diagram, the DEGs were involved in porphyrin and chlorophyll metabolism, flavonoid biosynthesis, metabolic pathways, circadian rhythm-plants, ubiquinone, and other terpenoid-quinone biosynthetic KEGG pathways in ST/CK. The DEGs in MST/ST were involved in glutathione metabolism and cutin, suberin, and wax biosynthesis (Figure 8). Studies have shown that porphyrin and chlorophyll metabolism help *Malus halliana* seedlings resist salt-alkali stress [31]. Glutathione (GSH) exists in the form of oxidized glutathione (GSSG) and reduced glutathione (GSH), which scavenge ROS. Nutrient stress can reduce the accumulation of GSH and GSSG and interfere with glutathione metabolism [32]. In this study, adding exogenous melatonin promoted the absorption and utilization of nutrients, thus regulating the GSH cycle to reduce adverse reactions under stress conditions. In addition, most of these genes are involved in chlorophyll synthesis, the stress response, metal-ion transport, and phosphate balance. Among them, some metal-ion-transport-related genes were downregulated by nutrient stress, while exogenous melatonin significantly alleviated the downregulation of some metal-ion-transport-related genes (Appendix A). K deficiency regulates gene expression less than N or P deficiency [33]. However, few studies have examined the transcriptome in plants under nutrient stress. Our results show that the effect of nutrient stress on the transcriptome was not as great as that of a single mineral.

Nutrient stress affects plant metabolism [34]. For example, when N is scarce, amino-acid levels in plant leaves significantly decrease [35]. Most amino acids and their derivative organic acids and flavonoids change significantly in the root system of P-deficient apple seedlings [5], and phosphorylated metabolites in roots decrease significantly [20]. Remodeling of the metabolome under nutrient stress largely reflects the response and defense of the plant’s body to stress. In this study, nutrient stress resulted in a decline in concentration of amino acids, organic acids, and phenolic acids. Notably, oxalic acid, L-ascorbic acid, anthocyanins (cyanidin-3-O-galactoside), lignans (lirioresinol A and syringaresinol), and melatonin significantly increased after exogenous melatonin was applied to plants under nutrient stress (Appendix A). Oxalic acid plays a positive role in regulating metal stress, ion stress, and disease from insect pests [36]. It plays an important role in enhancing resistance to smut in *Triticum aestivum* [37]. Plants have developed a variety of strategies to ameliorate the harmful effects of stress. The production of additional antioxidant protectants is one important strategy. Ascorbic acid and anthocyanins are effective antioxidants and are present in almost all photosynthetic eukaryotes [38]. Anthocyanin metabolism is involved in freezing and drought resistance in plants [39,40]. Ascorbic acid also enhances the resistance to drought stress [41]. In addition, lignin is the main component of the cell walls of vascular plants; it promotes lignification and enhances the ability of plants to resist stress, such as enhancing the resistance of *Zea mays* to leaf blight and gray spot [32,42]. Melatonin acts as the first line of defense against oxidative stress in the internal and external environments. The results of this experiment show that when exogenous melatonin is applied to apple plants, the leaves metabolize melatonin to help the plants resist oxidative damage caused by the nutrient stress. The increase in these beneficial metabolites may be the key way melatonin helps apple plants resist nutrient stress and promotes the absorption and utilization of nutrients in plants under nutrient stress.

## 4. Materials and Methods

### 4.1. Plant Materials and Growth Conditions

*Malus hupehensis* from triploid and apomixis-type seeds were gathered from Pingyi (35°07′ N, 117°25′ E) in Shandong, China. The experiments were carried out at Hebei Agricultural University located in Baoding (38°23′ N, 115°28′ E), Hebei, China. The seeds were treated with low-temperature stratification for about 30 days in early February 2021. Seeds hidden in the sand were sown in early March 2021 and watered regularly to ensure growth.

### 4.2. Experimental Design

Seedlings of a similar size (7–8 leaves, about 8 cm high) were transferred to hydroponic pots filled with 10 L 1/2 strength Hoagland nutrient solution. The pots were left in hydroponic culture for 15 days to adapt to the new growing environment. The nutrient solution was ventilated through an air pump to prevent hypoxic necrosis of the roots. The pH of the nutrient solution was adjusted to 6.0 ± 0.1 with H_3_PO_4_ and refreshed every five days. On day 12 of pre-culture, the materials were evenly divided into two groups; one group was the control group, and the other group was treated with melatonin (0.1 μM). After the end of pre-culture, the defective seedlings were removed and treated for nutrient deficiency. The experimental materials were divided into four new experimental groups: 1/2 Hoagland nutrient solution (CK); 1/2 Hoagland nutrient solution + 0.1 µM melatonin (MCK), 1/20 Hoagland nutrient solution (ST), and 1/20 Hoagland nutrient solution + 0.1 µM melatonin (MST). The experiment lasted 20 days with three replicates in each treatment and 80 plants in each group. After 20 days, a total of 10 seedlings were selected and 3–4 uniform, healthy, mature leaves from each plant were harvested and frozen in liquid nitrogen for further analysis.

### 4.3. Measurement of Growth Characteristics and Root Architecture

After 20 days of hydroponic treatment, six uniform and healthy seedlings were selected for each treatment and final PL and TD were measured. Then, the whole plant was divided into roots, stems, and leaves; washed with deionized water to remove all impurities; and dried with a paper towel. The plants were fixed at 105 °C for 30 min, then dried at 65 °C for three days. TDW and RSR of the plants were determined, as described by Liang et al. [8]. RGR was calculated with the formula, [(lnDW2 − lnDW1)/(T2 − T1)], where DW1 is the dry weight of the plant on day 0 (T1), and DW2 is the dry weight of the plant on day 20 (T2) [43]. WinRHIZO^®^ image analysis software (V4.1C; Regent Instruments, Quebec City, QC, Canada) was used to analyze total root length, diameter, surface area, volume, tips, and forks [6].

### 4.4. Determination of TCC, TTC, REL, MDA, and Antioxidant Enzyme Activity

TCC was determined via 80% acetone calorimetry, as described by Liang et al. [8]. Root activity was measured using the TTC method [2]. REL was measured with a thundermagnetic DDS-307 conductivity meter [44]. MDA content was determined using the thiobarbituric acid method [14]. Fresh leaves (0.2 g) were ground with 10% trichloroacetic acid (1.6 mL), centrifuged, and 1.5 mL 0.67% thiobarbituric acid was added to 1.5 mL supernatant. After boiling in water for 30 min, the absorbance values at 450 nm, 532 nm, and 600 nm were determined via spectrophotometry.

Fresh leaves (0.2 g) were washed and placed in a pre-cooled mortar. A 1.6 mL aliquot of 0.05 mmol phosphoric acid buffer (pH = 7.8) was added and the leaves were ground. After centrifugation, SOD, POD, and CAT of leaves were measured. SOD was determined using the photochemical nitroblue tetrazolium method [14]. POD was determined using the guaiacol method [14]. CAT was determined using the ultraviolet absorption method [14]. APX was determined according to the kit instructions (Suzhou Keming Biotechnology Co., Ltd., Suzhou, China) by adding 1 mL reagent to 0.1 g fresh leaves followed by grinding on ice. After centrifugation, APX was determined by spectrophotometry [45].

### 4.5. Determination of the Stomatal Configuration

Ten leaves were collected from the same position per treatment group. The leaves were cut from both sides of the main veins with a knife (5 mm × 5 mm sample squares) and quickly placed in 4% glutaraldehyde prepared with 0.1 mm phosphate buffer (PBS, pH 6.8) for overnight fixation; twenty cubed leaves were obtained for each treatment. The leaves were rinsed, dried, and vacuum gilded. An S-4800 microscope (Hitachi Ltd., Tokyo, Japan) was used for scanning electron microscopy (SEM). Stomatal length and width, stomatal density, and stomatal opening were analyzed with ImageJ software (25 random sections) [46].

### 4.6. Determination of Minerals

Nine uniform and healthy seedlings were selected for each treatment. The dried root, stem, and leaf samples were ground and mixed, sieved, and reserved. A sample (0.3 g) was accurately weighed and placed in 100 mL digestion tubes, and 5 mL concentrated sulfuric acid was added to determine the N concentration. Then the tube was heated at 220 °C for 4 h in an electric digestion furnace (Multiwave PRO; Anton-Paar GmbH, Graz, Austria). Ten drops of hydrogen peroxide were added every 30 min until the solution was clear and transparent. After cooling, deionized water was added to the digestion tube at a constant volume. The 1 mL solution was diluted to 5 mL, and the minerals were determined using a segmented flow analyzer (AA3; SEAL Analytical, Norderstedt, Germany). A 0.15 g sample was accurately weighed and added to a digestive tube to determine K, Ca, and Mg concentrations. Then, 5 mL concentrated nitric acid and 1 mL perchloric acid were added. The digestion was completed in an electric digestion furnace, and inductively coupled plasma source mass spectrometry (ICAP Q; Thermo Fisher Scientific, Waltham, MA, USA) was performed after a constant volume was attained. The absorption, transfer, and distribution of the minerals were determined following the methods of Liang et al. [19].

### 4.7. Extracting RNA and Transcriptome Sequencing

The transcriptome analysis was performed on nine leaf samples, including the CK, ST, and MST. There were three replicates for each treatment, and leaves of 30 seedlings were collected from the same position per replicate (2–3 uniform, healthy, mature leaves from each plant). The total RNA of the apple leaves was separated using TRIzol reagent (Invitrogen, Carlsbad, CA, USA) for transcriptome analysis. After extraction of total RNA, Illumina RNA-Seq was performed by Metware Biotechnology Co. Ltd. (Wuhan, China). Purified RNA (1 µg each sample) was reverse transcribed to first-strand cDNA using the cDNA Reverse Transcription Kit (PrimeScript^TM^ RT Master Mix, Takara Bio, Ohtsu, Japan) according to the manufacturer’s instructions. The raw reads were transformed from the raw sequencing image data using CASAVA base recognition. The adapter sequences were cut, and low-quality reads with ≥ 5 uncertain bases or with more than 50% 4 Qphred ≤ 20 bases were removed using fastp to obtain the high-quality data. The GC content of the clean reads was calculated. The Q20 and Q30 values were also determined by FastQC to evaluate base quality. Fragments per kilobase of transcript per million fragments mapped was calculated as an indicator to measure the transcripts or DEGs. DESeq2 is a suitable method for differential expression analysis between sample groups with biological replicates to obtain DEG sets between two biological conditions [47]. The conditions for allogeneic screening were |log2 (fold change)| ≥ 1 and a false discovery rate < 0.05. The DEGs were analyzed using gene ontology (GO) and the Kyoto Encyclopedia of Genes and Genomes (KEGG) tools [48].

### 4.8. qRT-PCR Validation

Total RNA was extracted from each treated leaf using the M5 Plant RNeasy Complex Mini Kit (Mei5 Biotechnology Co., Ltd., Beijing, China), as directed by the manufacturer. The inverse transcription was carried out using the UEIrisIIRT-PCR System for First-Strand cDNA Synthesis system (Suzhou US Everbright, Inc., Suzhou, China). The primers for all genes are shown in Appendix A. Three replicates were set for each treatment, and the 2^−∆∆Ct^ method was used to analyze the normalized expression of each sample [49].

### 4.9. Metabolite Analysis

The metabolome analysis was performed on nine leaf samples, including the CK, ST, and MST. There were three replicates for each treatment, and leaves of 30 seedlings were collected from the same position per replicate (2–3 uniform, healthy, mature leaves from each plant). Sample preparation and extraction, metabolome profiling, and data analysis were performed according to the standard procedures of Wuhan MetWare Biotechnology Co., Ltd. (Wuhan, China) (www.metware.cn, accessed on 6 December 2020). The sample extracts were analyzed using an ultrahigh-performance liquid chromatography-electrospray ionization tandem mass spectrometry system (HPLC, Shimadzu Nexera X2, Kyoto, Japan, www.shimadzu.com.cn/, accessed on 6 December 2020; MS, Applied Biosystems 4500 Q TRAP, Carlsbad, CA, USA, www.appliedbiosystems.com.cn/, accessed on 6 December 2020). The metabolite data were log2-transformed to improve normality for the statistical analysis and were normalized. Principal component analysis (PCA) was carried out to preliminarily understand the overall metabolic differences among the samples in each group and the degree of variation among the samples within the group. Metabolites with variable importance in projection (VIP) ≥ 1.0, |log_2_(fold change)| ≥ 1 were defined as significantly changed metabolites (SCMs). A principal component analysis (PCA) of the SCMs was performed using R [48].

### 4.10. Statistical Analysis

Data were analyzed using IBM SPSS Statistics 20 software (IBM Corp, Armonk, NY, USA), and charts were plotted using SigmaPlot 10.0 software (Systat Software, Inc., San Jose, CA, USA). Univariate analysis of variance was used to compare the mean values between treatments, and Tukey’s multi-range test was used to detect differences between treatments. A *p*-value < 0.05 was considered significant.

## 5. Conclusions

We conducted physiological, ionomic, transcriptomic, and metabolomic analyses of apple plants responding to nutrient stress after applying exogenous melatonin. Our results demonstrate that nutrient stress significantly affected the growth, stomatal structure, physiological characteristics, and antioxidant enzyme activities of the apple seedlings. Exogenous melatonin significantly alleviated the damage caused by nutrient stress, and the plants showed stronger tolerance and adaptability. The absorption, distribution, and utilization efficiencies of macroelements and microelements were different in apple plants under different conditions. Exogenous melatonin significantly promoted the absorption and utilization of minerals in apple plants and changed their distributions. The transcriptome and metabolome were compared and analyzed. Apple seedlings responded to nutrient stress by regulating the GSH pathway, upregulating some metal-ion-transporter genes and increasing beneficial metabolites. These results provide a basis for the use of melatonin to alleviate nutrient deficiencies during agricultural production.

## Figures and Tables

**Figure 1 ijms-23-09855-f001:**
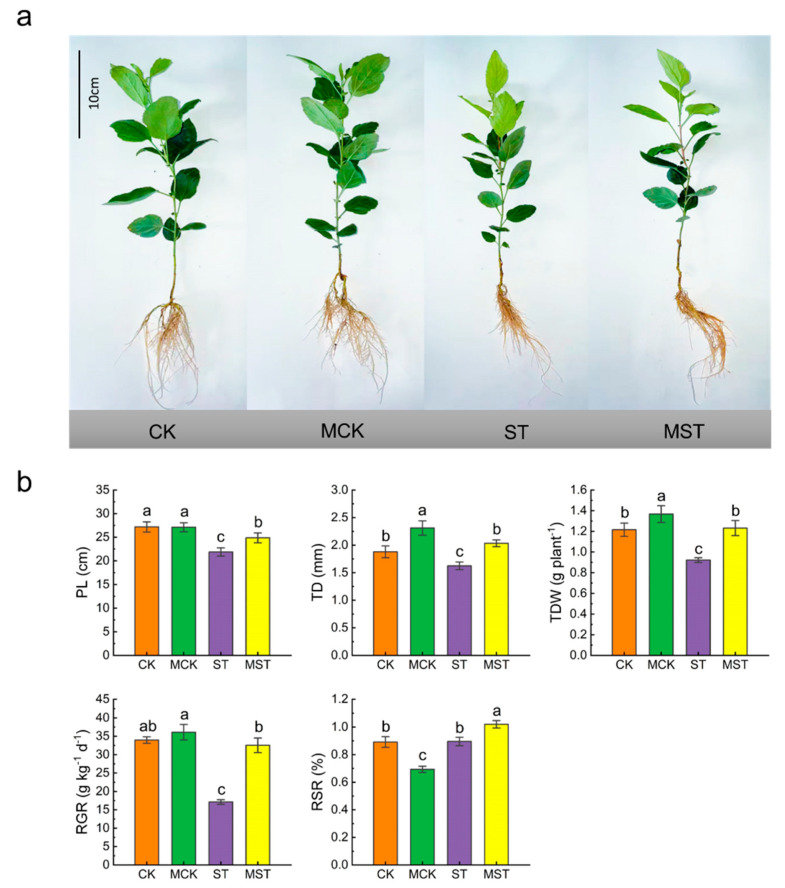
Effects of melatonin on plant growth under nutrient stress: (**a**) plant phenotypic, (**b**) plant length (PL), trunk diameter (TD), total dry weight (TDW), relative growth rate (RGR), and root/stem ratio (RSR). Data are means ± SD of four replicate samples. Values labelled with different letters are significantly different by Tukey’s multiple range tests (*p* < 0.05). CK, 1/2 Hoagland nutrient solution; MCK, 1/2 Hoagland nutrient solution with 0.1 μmol/L melatonin; ST, 1/20 Hoagland nutrient solution; MST, 1/20 Hoagland nutrient solution with 0.1 μmol/L melatonin.

**Figure 2 ijms-23-09855-f002:**
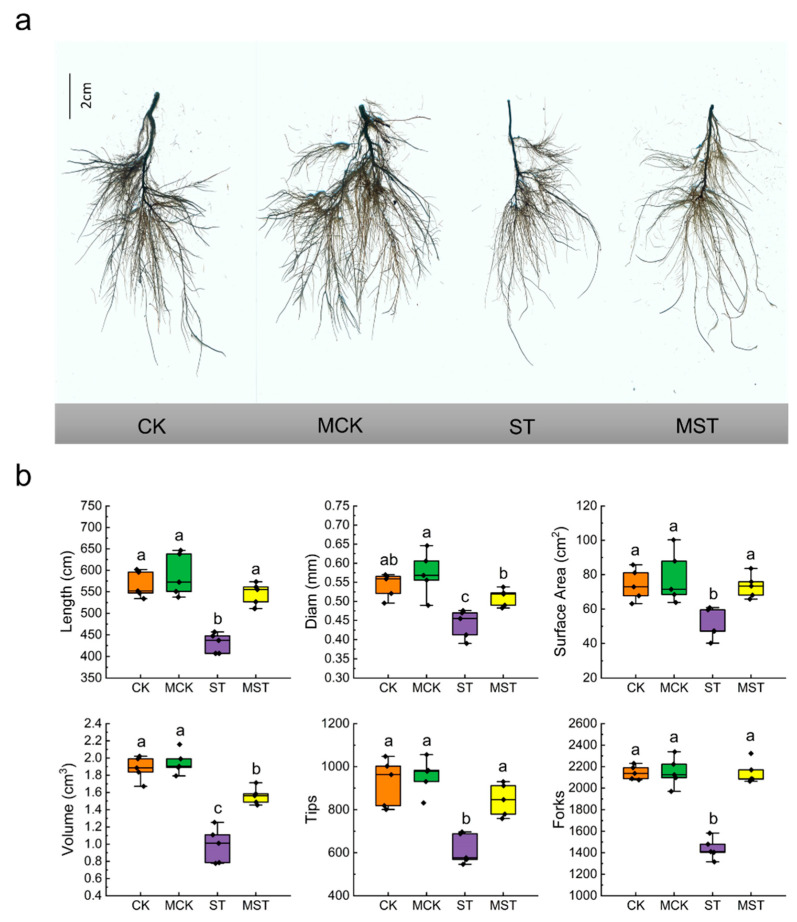
Root architecture under different treatments: (**a**) root architecture scan, (**b**) the lengths, diams, surface areas, volume, tips, and forks of roots. Data are means ± SD of five replicate samples. Values labelled with different letters are significantly different by Tukey’s multiple range tests (*p* < 0.05). CK, 1/2 Hoagland nutrient solution; MCK, 1/2 Hoagland nutrient solution with 0.1 μmol/L melatonin; ST, 1/20 Hoagland nutrient solution; MST, 1/20 Hoagland nutrient solution with 0.1 μmol/L melatonin.

**Figure 3 ijms-23-09855-f003:**
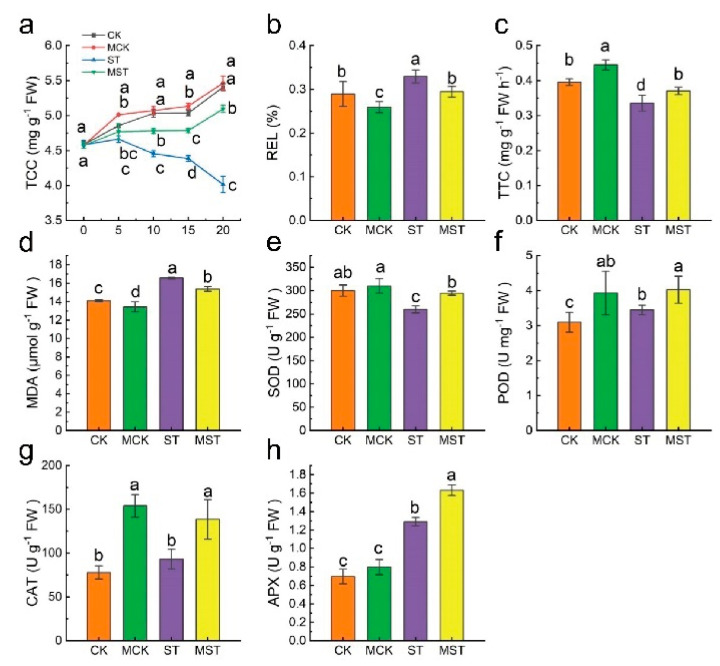
Effects of different treatments on total chlorophyll content (TCC), relative electrical conductivity (REL), the root activity (TTC), malondialdehyde (MDA) content, and antioxidant enzyme activities. (**a**) TCC, (**b**) REL, (**c**) TTC, (**d**) MDA content, (**e**) superoxide dismutase (SOD) activity, (**f**) peroxidase (POD) activity, (**g**) catalase (CAT) activity, and (**h**) ascorbate peroxidase (APX) activity. Data are means ± SD of three replicate samples. Values labelled with different letters are significantly different by Tukey’s multiple range tests (*p* < 0.05). CK, 1/2 Hoagland nutrient solution; MCK, 1/2 Hoagland nutrient solution with 0.1 μmol/L melatonin; ST, 1/20 Hoagland nutrient solution; MST, 1/20 Hoagland nutrient solution with 0.1 μmol/L melatonin.

**Figure 4 ijms-23-09855-f004:**
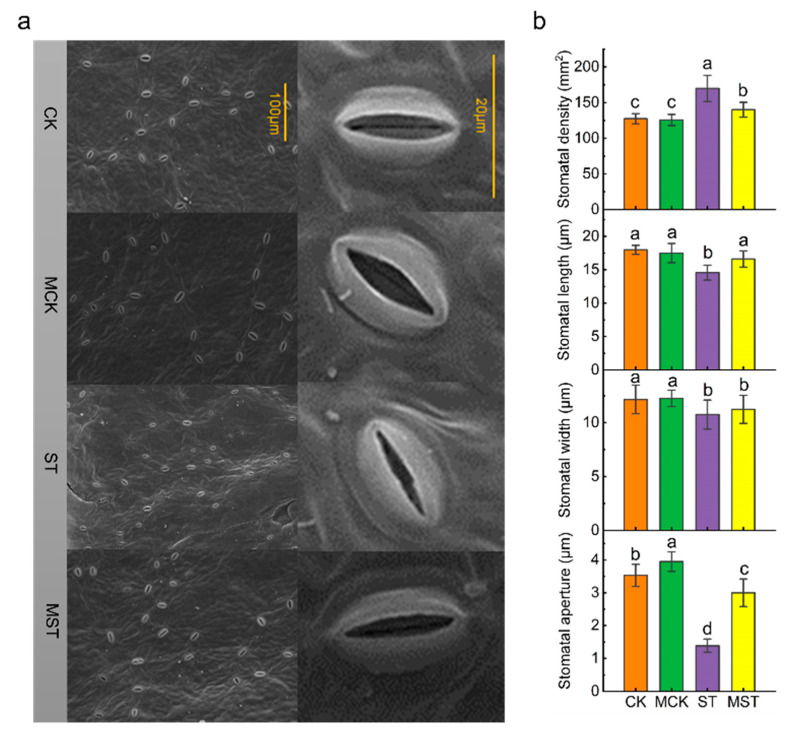
Melatonin regulates stomatal development and opening of apple leaves under nutrient deprivation: (**a**) electron microscopy of stomata, (**b**) stomatal density, stomatal length, stomatal width and stomatal aperture under different treatment conditions. Data are means ± SD of five replicate samples. Values labelled with different letters are significantly different by Tukey’s multiple range tests (*p* < 0.05). CK, 1/2 Hoagland nutrient solution; MCK, 1/2 Hoagland nutrient solution with 0.1 μmol/L melatonin; ST, 1/20 Hoagland nutrient solution; MST, 1/20 Hoagland nutrient solution with 0.1 μmol/L melatonin.

**Figure 5 ijms-23-09855-f005:**
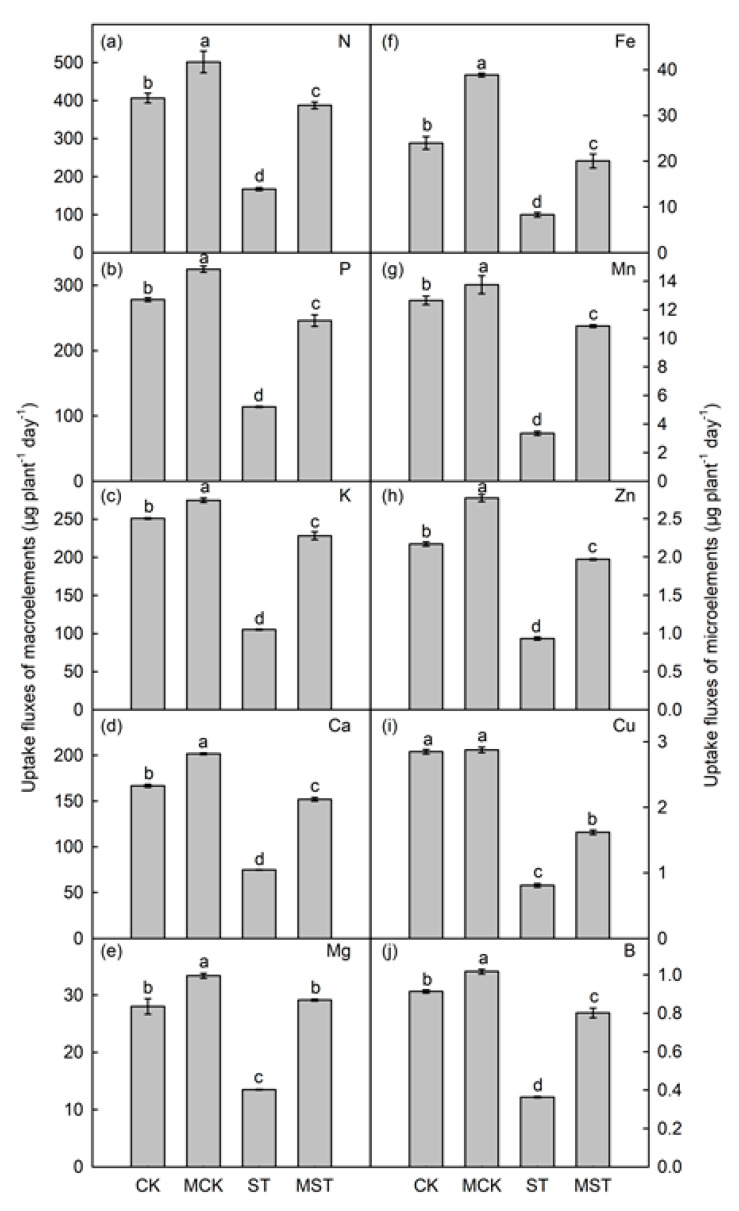
Effects of melatonin treatment on uptake fluxes of macro- and microelements (μg plant^−1^ day^−1^) in *Malus hupehensis*. (**a**–**e**) Uptake fluxes of macronutrients (N, P, K, Ca and Mg) and (**f**–**j**) micronutrients (Fe, Mn, Zn, Cu and B). Data are means ± SD of four replicate samples. Values labelled with different letters are significantly different by Tukey’s multiple range tests (*p* < 0.05). CK, 1/2 Hoagland nutrient solution; MCK, 1/2 Hoagland nutrient solution with 0.1 μmol/L melatonin; ST, 1/20 Hoagland nutrient solution; MST, 1/20 Hoagland nutrient solution with 0.1 μmol/L melatonin.

**Figure 6 ijms-23-09855-f006:**
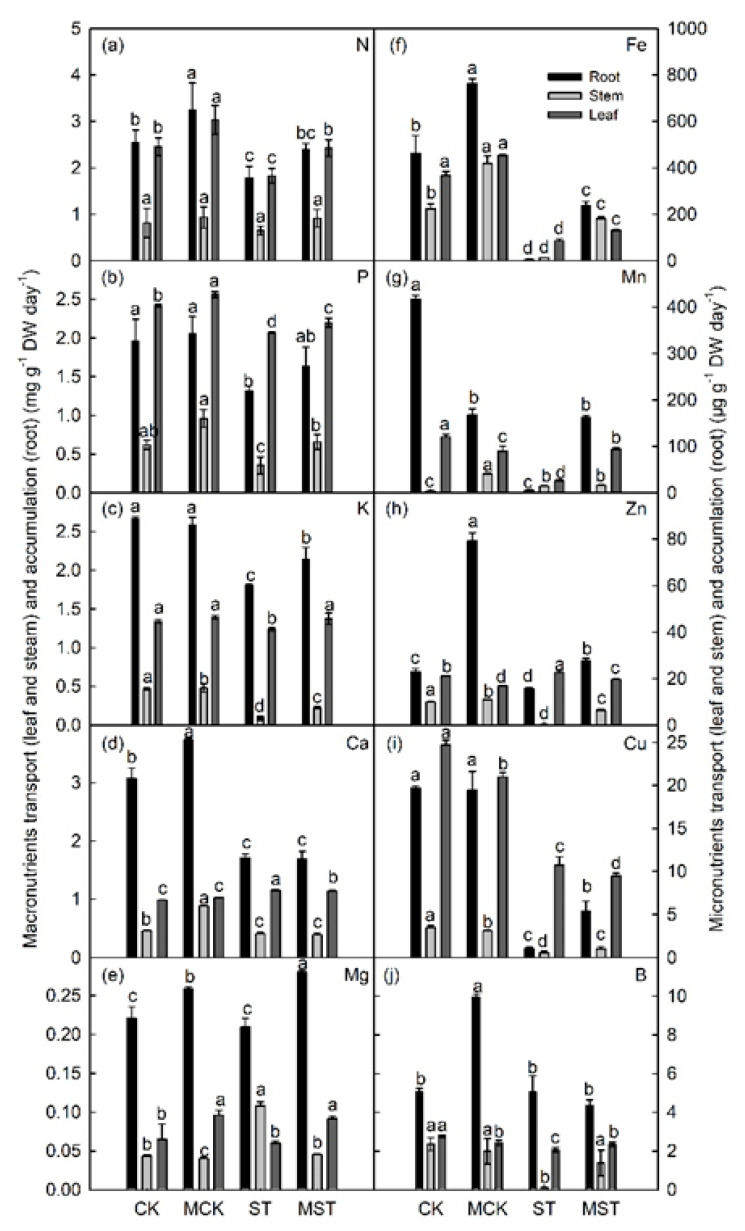
Nutrient transport and accumulation in *Malus hupehensis* plants under different treatment conditions: (**a**–**e**) macronutrients (N, P, K, Ca, and Mg) transport (leaf and stem) and accumulation (root), and (**f**–**j**) micronutrients (Fe, Mn, Zn, Cu, and B) transport (leaf and stem) and accumulation under different treatment conditions. Data are means ± SD of four replicate samples. Values labelled with different letters are significantly different by Tukey’s multiple range tests (*p* < 0.05). CK, 1/2 Hoagland nutrient solution; MCK, 1/2 Hoagland nutrient solution with 0.1 μmol/L melatonin; ST, 1/20 Hoagland nutrient solution; MST, 1/20 Hoagland nutrient solution with 0.1 μmol/L melatonin.

**Figure 7 ijms-23-09855-f007:**
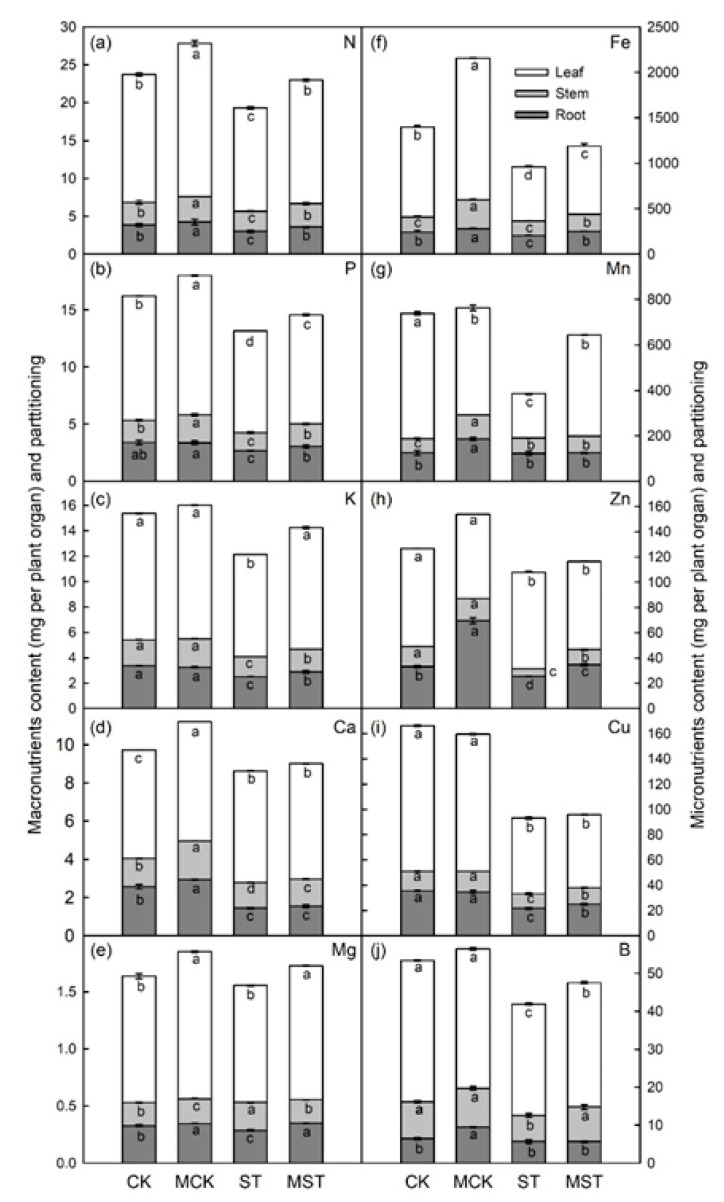
Melatonin regulates the content and partition of elements in whole plants under nutrient deprivation: (**a**–**e**) macronutrients (N, P, K, Ca, and Mg) content and partition, and (**f**–**j**) micronutrients (Fe, Mn, Zn, Cu and B) content and partition in whole plants under different treatment conditions. Data are means ± SD of four replicate samples. Values labelled with different letters are significantly different by Tukey’s multiple range tests (*p* < 0.05). CK, 1/2 Hoagland nutrient solution; MCK, 1/2 Hoagland nutrient solution with 0.1 μmol/L melatonin; ST, 1/20 Hoagland nutrient solution; MST, 1/20 Hoagland nutrient solution with 0.1 μmol/L melatonin.

**Figure 8 ijms-23-09855-f008:**
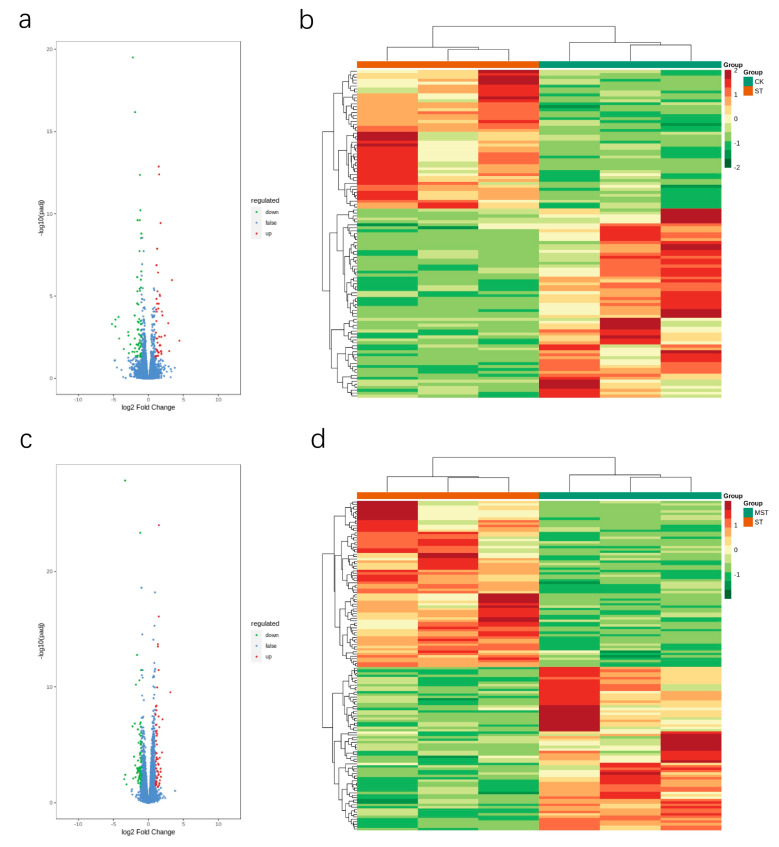
Transcriptome statistics of DEGs under different treatment conditions: (**a**,**b**) the volcano pot and cluster heat map of differently expressed genes in ST/CK; (**c**,**d**) the volcano plot and cluster heat map of differently expressed genes in MST/ST. The abscissa indicates the sample name and hierarchical clustering results, and the ordinate indicates differently expressed genes and hierarchical clustering results.

**Figure 9 ijms-23-09855-f009:**
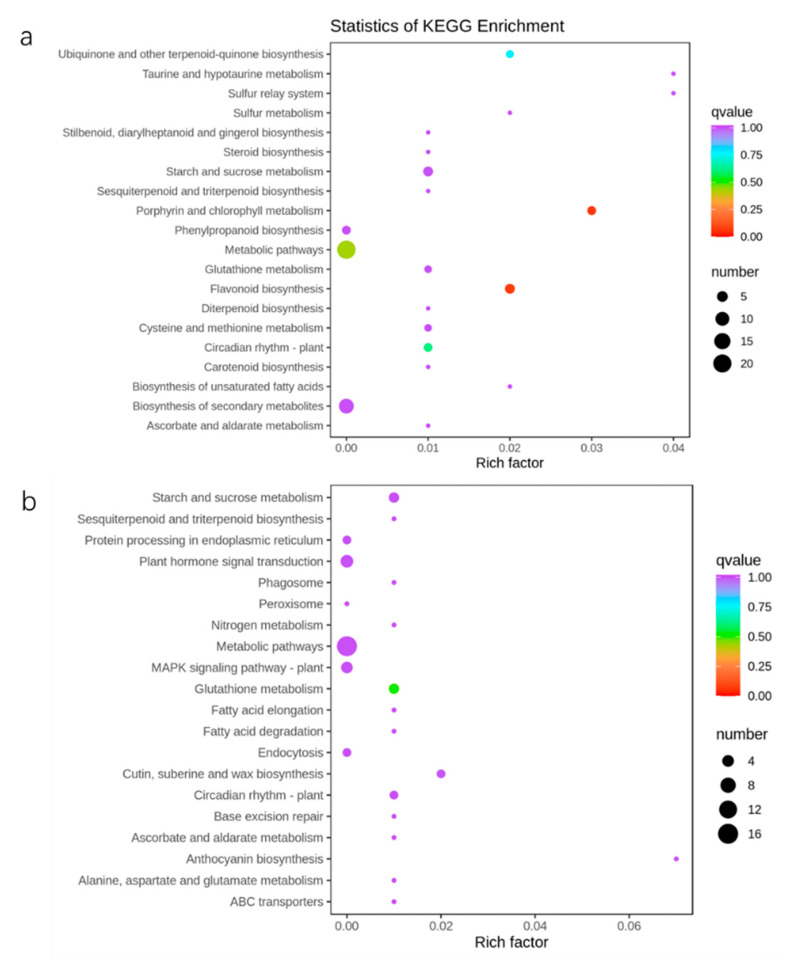
Top 20 KEGG enrichment analyses of differently expressed genes under different conditions in apple leaves: (**a**) statistics of KEGG enrichment in ST/CK; (**b**) statistics of KEGG enrichment in MST/ST. Each circle represents a KEGG pathway, the ordinate represents the pathway name, and the abscissa is the enrichment factor. The larger the enrichment factor is, the greater the degree of enrichment is. The circle color represents q-value, the smaller the q-value is, the more reliable the enrichment significance is. The size of the circle indicates the number of genes enriched in the pathway, and the larger the circle, the more abundant the genes is.

**Figure 10 ijms-23-09855-f010:**
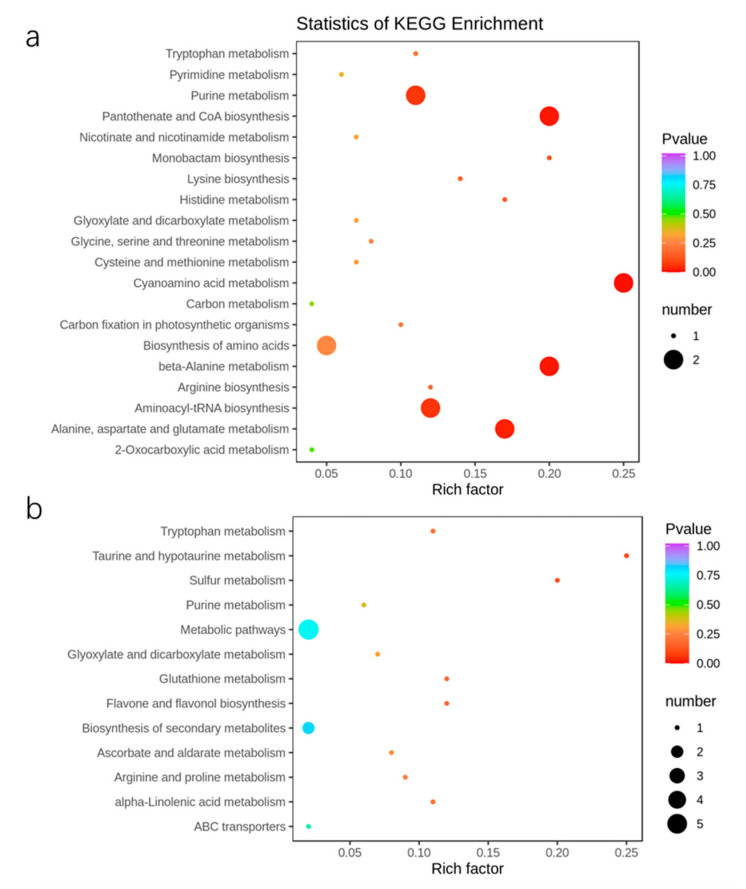
Top 20 KEGG enrichment analyses of different metabolites under different conditions in apple leaves: (**a**) statistics of KEGG enrichment in ST/CK; (**b**) statistics of KEGG enrichment in MST/ST. Each circle represents a KEGG pathway, the ordinate represents the pathway name, and the abscissa is the enrichment factor. The larger the enrichment factor is, the greater the degree of enrichment is. The circle color represents *p*-value; the smaller the *p*-value is, the more reliable the enrichment significance is. The size of the circle indicates the number of metabolites enriched in the pathway, and the larger the circle, the more abundant the metabolites is.

## Data Availability

The raw sequence data were deposited in the NCBI Sequence Read Archive (SRA) database under accession number PRJNA874497.

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
