# Peer review of "Ionomic Combined with Transcriptomic and Metabolomic Analyses to Explore the Mechanism Underlying the Effect of Melatonin in Relieving Nutrient Stress in Apple"

_ijms, 2022, doi:10.3390/ijms23179855_

Round 1

Reviewer 1 Report

The manuscript presents the data on the role of melatonin on the nutrient stress.  Lacks new information considering the authors's previous publication  (Du et al., 2022) as well as publications by other.

Presented transcriptomic and metabolic analysis is rather limited in scope. Besides these concerns, the additional concerns are:

1. Figure 1a and 2A can benefit with presentation of a reference scale to highlight the differences in plant as well as root length and width. Although the accompanying panel indicate the lengths, presentation of reference scale in Fig 1A is important.

2. Statistical significance needs to be calculated and presented for all the bar diagrams (Fig. 1 - Fig. 6).

3. Table 1 caption states "Effects of Dopamine treatment ....". Dopamine or melatonin? 

4. Regarding the transcriptomic analysis, The significance of the differentially expressed genes are not presented.  Differentially expressed genes need to be validated and discussed. Mere presentation of the heat map and the discussion of the KEGG analysis do not add any significance. The number of samples analyzed need to be presented.

5. The metabaolomic data is presented without highlighting the significance of the findings. What is the significance of the differential metabolite and the consensus metabolites. Similar concerns on the number analyzed samples and validation need to be presented.

6. The citations should include other studies on the effect of melatonin on apple plants including those of Verde et al., 2022 and Mao et al., 2020.

7. Figure 2 of the present study is similar to the one presented by the authors in the previous publication (Du et al., 2022).

Author Response

Responses to Reviewer #1

Thank you very much for the opportunity that you allowing us to revise our manuscript. We are grateful to your valuable and constructive suggestions which would further improve the quality of our manuscript. As for your question about the innovation of the manuscript, we are also very concerned, and here carefully list the highlights of this manuscript: First of all, the Du et al (2022) study you mentioned is for N deficiency, while our study is multi-nutrient deficiency. This is not a simple repetition, and studies on the regulatory mechanisms of melatonin in apple plants under nutrient stress conditions have not yet been carried out. Since this stress may cause complex changes in physiological and biochemical activities in plants, we further analyze the transcriptional metabolism of melatonin on the basis of determining its ability to alleviate nutrient stress through phenotypic indicators and element content. Trying to initially locate the key genes and metabolic substances in response to melatonin. These results will provide the basis for further in-depth study in the future. We also read the comments thoroughly and have modified the manuscript according to your comments. Changes and amendments are marked up using the ‘Track Changes’ in the revised manuscript. Point by point responses to comments are listed below.

References:

Du, P. H., Yin, B. Y., Cao, Y., Han, R. X., Ji, J. H., He, X. L., Liang, B. W., and Xu, J. Z. (2022). Beneficial effects of exogenous melatonin and dopamine on low nitrate stress in Malus hupehensis. Frontiers in Plant Science 12.

Comment 1: Figure 1a and 2a can benefit with presentation of a reference scale to highlight the differences in plant as well as root length and width. Although the accompanying panel indicate the lengths, presentation of reference scale in Fig 1a is important.

Answer 1: Thanks for your comment. Your comments have been of great help in improving the accuracy of our images and have also provided guidance for our future work, which we will adopt and revise with an open mind. The reference scales in Figure 1a and Figure 2a have been supplemented. We used to put a scale next to the photo, but we ignored its importance. In addition, the root-scanned instrument in Figure 2 has a scale, both of which provide an accurate reference for the annotation of the reference scale in the figure.

Comment 2: Statistical significance needs to be calculated and presented for all the bar diagrams (Fig. 1 - Fig. 6).

Answer 2: Thanks for your comment. The significance calculation between the four treatments in the bar diagrams of our Figs. 1 to 6 are calculated by Tukey's range test. Different letters indicate significant differences among the four treatments as defined by Tukey's test (P < 0.05). Tukey's range test is a single-step multiple comparison procedure and statistical test. It can be used to find means that are significantly different from each other. Tukey's test is based on a formula very similar to that of the t-test. In fact, Tukey's test is essentially a t-test, except that it corrects for family-wise error rate. However, we noticed that Fig. 3a was not marked with significance, and we have added letter markers.

Comment 3: Table 1 caption states "Effects of Dopamine treatment ....". Dopamine or melatonin?

Answer 3: Thanks for your comment. Due to our carelessness, we would like to express our sincerest apologies for the trouble caused by your understanding. This manuscript studies melatonin without reference to dopamine, and we will revise it and review the full text carefully. The label in our Table 1 should be changed to “Effects of melatonin treatment on uptake fluxes of macro- and microelements (μg plant-1 day-1) in Malus hupehensis.” However, according to the second reviewer's opinion, we changed this table to Figure 5.

Comment 4: Regarding the transcriptomic analysis, The significance of the differentially expressed genes are not presented.  Differentially expressed genes need to be validated and discussed. Mere presentation of the heat map and the discussion of the KEGG analysis do not add any significance. The number of samples analyzed need to be presented.

Answer 4: Thanks for your comment. The transcriptome analysis is performed on nine leaf samples, including the CK, ST, and MST with three replicates per treatment.

In addition, we have performed validation on the transcriptome data. The supplemental content is as follows:

  1. Materials and Methods

4.9 qRT-PCR validation

Total RNA was extracted from each treated leaf using the M5 Plant RNeasy Complex Mini Kit (Mei5 Biotechnology Co., Ltd., Beijing, China) as directed by the manufacturer. The inverse transcription was carried out using the UEIrisIIRT-PCR System for First-Strand cDNA Synthesis system (Suzhou US Everbright, Inc., Suzhou, China). The primers for all genes were shown in Table S9. Three replicates were set for each treatment, and used the 2−∆∆Ct method to analyze the normalized expression of each sample.

(gene-MD05G1178500, gene-MD10G1227800, gene-MD04G1117800, gene-MD01G1089600, gene-MD12G1125600, gene-MD03G1251400, gene-MD05G1132400, gene-MD04G1164500, gene-MD10G1024000, gene-MD12G1178200 and gene-MD12G1178500).

  1. Results

2.6 Validation of DEGs by qRT-PCR

To confirm the RNA-seq results, eleven DEGs that had different roles in plant leaves were selected. The RT-qPCR and RNA-Seq results were consistent for all of the ten validated genes (Figure S5), indicating that reliable RNA-seq data were obtained from the samples.

With the development of gene sequencing related technologies and the reduction of sequencing costs, RNA-seq has become the main method of transcriptome research with its advantages of high throughput, high sensitivity and wide application range. A total of 9 samples is sequenced for transcriptome sequencing, yielding a total of 65.88 Gb clean data, each sample with a Clean Data of 6 Gb and a Q30 base percentage of 94% or more. Based on the comparison results, variable splicing prediction analysis, gene structure optimization analysis and new gene exploration are performed. Based on the comparison results, the gene expression amount is analyzed. Differentially expressed genes are identified according to the amount of gene expression in different samples, and their functional annotation and enrichment analysis are analyzed.

The number of fragments in a transcript is related to the amount of sequencing data (or Mapp Data), transcript length, transcript expression level, in order for the number of fragments to truly reflect the transcript expression level, the number of Mapp Reads in the sample and the length of transcripts need to be normalized. Using FPKM (Fragments Per Kilobase of transcript per Million fragments mapped) as a measure of transcript or gene expression levels, the FPKM calculation formula is as follows:

Mapped fragments of transctipt represents the number of fragments compared to a transcription book, that is, the number of double-ended Reads, and Total Count of mapped fragments (Millions) represents the total number of fragments compared to the transcription template, in units of 10^6; Length of transcript (kb): Transcript length, in 10^3 bases.

For samples with biological replicates, DESeq2 (Love, Huber, and Anders 2014; Varet et al. 2016) suitable for differential expression analysis between sample groups, obtaining differentially expressed gene sets between two biological conditions. DESeq2 requires unsorted reads count data for the input gene, not RPKM, FPKM, etc.

The reads count of genes is implemented using featureCounts (Liao, Smyth, and Shi 2014). After the difference analysis, the Benjamini-Hochberg method also needs to be used to correct the hypothesis test probability (P value) for multiple hypothesis testing to obtain the False Discovery Rate (FDR). The differential gene filter is | log2Fold Change| > = 1 and the FDR < 0.05.

After using DESeq2 to complete the analysis of differentially expressed genes, the total number of different genes, the number of upregulated genes, and the number of downregulated genes in each group were counted. According to these, we have screened the key differential genes that may be involved in melatonin regulation of apple plants to alleviate total deficiency stress, which has scientific significance and reference value.

References:

Liao, Y., Smyth, G. K., and Shi, W. (2014). featureCounts: an efficient general purpose program for assigning sequence reads to genomic features. Bioinformatics 30, 923-30.

Love, M. I., Huber, W., and Anders, S. (2014). Moderated estimation of fold change and dispersion for RNA-seq data with DESeq2. GENOME BIOLOGY 15.

Varet, H., Brillet-Gueguen, L., Coppee, J. Y., and Dillies, M. A. (2016). SARTools: A DESeq2- and EdgeR-Based R Pipeline for Comprehensive Differential Analysis of RNA-Seq Data. PLoS One 11, e0157022.

Comment 5: The metabaolomic data is presented without highlighting the significance of the findings. What is the significance of the differential metabolite and the consensus metabolites. Similar concerns on the number analyzed samples and validation need to be presented.

Answer 5: Thanks for your comment. Genes related to chlorophyll synthesis, the stress response, metal ion transport, and phosphate balance were changed at the same time in both treatments. The up-regulation of genes related to stress resistance and metal ion transport regulated by melatonin under nutritional stress will provide a basis for further research. Changes in these genes reinforce our need to explore downstream metabolites. Similarly, the metabolome analysis is performed on nine leaf samples, including the CK, ST, and MST with three replicates per treatment. Samples from all metabolomes correspond to samples from the transcriptome. This ensures the scientific nature of the joint analysis.

A total of 849 metabolites is detected based on UPLC-MS/MS detection platform and self-established database. Among them, the differential metabolites are shown in Table S5. The combination of chromatography and mass spectrometry realizes the whole process from the use of chromatography for substance separation to the use of mass spectrometry for substance identification. Ultra-high performance liquid chromatography-tandem mass spectrometry (UPLC-MS/MS) can accurately qualitatively and quantitatively.

Based on the OPLS-DA results, the Variable Importance in Projection (VIP) of the OPLS-DA model obtained from multivariate analysis could preliminarily screen out the differential metabolites between treatment. At the same time, p-value or fold change of univariate analysis could be used to further screen differential metabolites. Screening criteria were as follows:

  1. Metabolites with fold change≥2 and fold change≤0.5 are selected. Metabolites that differed more than 2-fold or less than 0.5 between the control and experimental groups are considered significant.
  2. Metabolites with VIP≥1 were selected. The VIP value represents the influential strength of the difference between groups of corresponding metabolites in the classification of samples in the model. Generally, metabolites with VIP≥1 are considered to be significantly different.

Comment 6: The citations should include other studies on the effect of melatonin on apple plants including those of Verde et al., 2022 and Mao et al., 2020.

Answer 6: Thanks for your comment. The references you provided have contributed greatly to improving the level of our articles, and we have carefully studied the contents of them. We have added this in the introduction to the article: “Previous studies have reported that melatonin can improve the formation of apple adventitious roots, accelerate the invisible propagation of apple rootstocks that are difficult to root [12], and improve the quality of apple fruit [13].”

References:

  1. Mao, J.P.; Niu, C.D.; Li, K.; Chen, S.Y.; Tahir, M.M.; Han, M.Y.; Zhang, D. Melatonin promotes adventitious root formation in apple by promoting the function of MdWOX11. BMC Plant Biol, 2020, 20, 536-536.
  2. Verde, A.; Míguez J.M.; Gallardo, M. Role of Melatonin in apple fruit during growth and ripening: possible interaction with ethylene. Plants-Basel. 2022, 11, 688-688.

Comment 7: Figure 2 of the present study is similar to the one presented by the authors in the previous publication (Du et al., 2022).

Answer 7: Thanks for your comment. Roots are the main organ for plants to obtain external nutrients, so root structure is a key morphological indicator of nutrient stress. The effect of nutrient stress and the promoting effect of melatonin on nutrient absorption and utilization could be judged by root configuration. Therefore, it is necessary in the manuscript. Although similar figure appeared in the article by Du et al (2022), first of all, our picture is the stress caused by the simultaneous lack of multiple nutrients, which is fundamentally different from the lack of a single element. Secondly, we use boxplots to better display the distribution of samples.

References:

Du, P. H., Yin, B. Y., Cao, Y., Han, R. X., Ji, J. H., He, X. L., Liang, B. W., and Xu, J. Z. (2022b). Beneficial effects of exogenous melatonin and dopamine on low nitrate stress in Malus hupehensis. Frontiers in Plant Science 12.

Reviewer 2 Report

Comments to manuscript ijms-1814653

Manuscript title:

Ionomic combined with transcriptomic and metabolomic analyses to explore the mechanism underlying the effect of melatonin in relieving nutrient stress in apple

The manuscript presents research on evaluation of melatonin impact on nutrient stress in apple with transcriptomic and metabolomic analyses.

The basic approaches are appropriate, the manuscript is well structured though it can be linguistically improved. The introduction better to be reorganized in order to provide more literature review on/around this topic. It needs to be structurally revised for better link between different topics. A logical connection between different parts of introduction seems missing. For instance, it might not be necessary to write about particular impact of different ions in physiological processes, but instead, the link between the subjects has to be better structured.

In the results, authors need more experimental support for some hypothesis e.g. line 109.  

Figure 4b, for the stomatal density seems, it seems there is no significant difference. that needs to be double checked!

Table 1, a figure may better shows the results!

The results of functional and KEGG analysis needs to precisely explained. Figure 8 legend is not informative.

There are several comments which comes in details below:

Line 26: what metabolome sequencing means!

Line 35: sentence revision

Line 64: what does excellent antioxidant means? Scientifically or physiologically excellent is not the proper word!

Lines 93-99: have to be revised

Line 109: what is the support for this hypothesis?

Line 192: Latin names in Italic

Line 216: DEGs, for the first mention in the text, has to be written in full words, abbreviated in brackets.

Line 222: were separate? What does it mean?

Line 515: check all latin names in italic

Author Response

Responses to Reviewer #2

Thank you very much for the opportunity that you allowing us to revise our manuscript. We are encouraged by the very positive comments and also grateful to your valuable and constructive suggestions which would further improve the quality of our manuscript. Based on your suggestion, we recognize the problem of the link between the various parts of the introduction. We have strengthened the coherence of the introductory paragraphs by modifying the content appropriately and adding transitional sentences. We also read the comments thoroughly and have modified the manuscript according to your comments. Changes and amendments are marked up using the ‘Track Changes’ in the revised manuscript. Point by point responses to comments are listed below.

Comment 1: In the results, authors need more experimental support for some hypothesis e.g. line 109. 

Answer 1: Thanks for your comment. The accuracy of the results narrative is important, and your comments are key to enhancing the scientific rigor of our articles.

As you said, although the root/stem ratio (RSR) of MST plants is significantly higher than that ST plants, this does not fully support subsequent hypotheses. We've deleted the follow-up notes.

Comment 2: Figure 4b, for the stomatal density seems, it seems there is no significant difference. that needs to be double checked!

Answer 2: Thanks for your comment. We carefully confirmed that the number of stomata in the four pictures in Figure 4a were 19(CK), 21(MCK), 31(ST) and 25(DST) respectively. The result of stomatal density is obtained by the formula of

And then we did the unit conversion. In addition, we have carefully checked the data for the stomatal in our article and use the Tukey's test (P < 0.05) to find that there are indeed significant differences.

Comment 3: Table 1, a figure may better shows the results!

Answer 3: Thanks for your comment. We agree with you very much and have carefully drawn Table 1 as a histogram, as shown in Figure 5.

Comment 4: The results of functional and KEGG analysis needs to precisely explained. Figure 8 legend is not informative.

Answer 4: Thanks for your comment. We have added a detailed explanation to the Figure 8 and 9. Add to the caption: “Each circle represents a KEGG pathway, the ordinate represents the pathway name, and the abscissa is the enrichment factor. The larger the enrichment factor is, the greater the degree of enrichment is. The circle color represents p-value, the smaller the p-value is, the more reliable the enrichment significance is. The size of circle indicates the number of metabolites enriched in the pathway, and the larger the circle, the more abundant the metabolites is.”

Comment 5: There are several comments which comes in details below:

Answer 5:

5.1 Line 26: what metabolome sequencing means!

Answer: Thanks for your comment. This is the ambiguity of our expression. Metabolomic detection does not involve sequence information in the process, so it should be called metabolomic detection. Therefore, the "Sequencing" in the original sentence is deleted.

5.2 Line 35: sentence revision

Answer: Thanks for your comment. We have rewritten it as “These differentially expressed genes and the increase in beneficial metabolites may explain how melatonin resists nutrient stress in plants.”

5.3 Line 64: what does excellent antioxidant means? Scientifically or physiologically excellent is not the proper word!

Answer: Thanks for your comment. The question you asked not only increased the academic rigor of this article, but also provided great help to the standardization of our future writing words. We have changed that sentence to “Melatonin is a pleiotropic molecule in plants that is synthesized by chloroplasts and mitochondria and exists in almost all plant tissues, where it has a strong antioxidant capacity and plays an important role in the regulation of stress resistance.”

5.4 Lines 93-99: have to be revised

Answer: Thanks for your comment. We have checked carefully and made the appropriate changes: “In this study, physiology, the ionome, the transcriptome, and the metabolome were used to analyze the response of apple plants to nutrient deficiency and explore the mechanism of tolerance and adaptability of melatonin in alleviating nutrient deficiency in apple seedling.”

5.5 Line 109: what is the support for this hypothesis?

Answer: Thank you for your questions. We have already answered it in question 1. We have deleted for hypotheses that were not adequately tested: “In addition, the root/stem ratio (RSR) of the MST plants was significantly higher than that of ST plants.”

5.6 Line 192: Latin names in Italic

Answer: Thanks for your comment. We have carefully checked the Latin names in the manuscript and corrected them to italics.

5.7 Line 216: DEGs, for the first mention in the manuscript, has to be written in full words, abbreviated in brackets.

Answer: Thanks for your comment. We've changed the first DEGs in this article to differentially expressed genes (DEGs).

5.8 Line 222: were separate? What does it mean?

Answer: Thanks for your comment. What we want to explain is that a clear distinguishing can occur between the three treatments. The problem of irregular wording has been revised in the manuscript. Principal component analysis (PCA) can preliminarily understand the overall metabolic difference between each group and the degree of variability between samples. We have checked carefully and made the appropriate changes: “The results showed an obvious difference among CK, ST, and MST (Figure S1).”

5.9 Line 515: check all latin names in italic

Answer: Thanks for your comment. We apologize to you for the repetition of this issue in the manuscript. We repeatedly proofread the Latin italics in the manuscript to make sure to correct the error. We have checked carefully and made the appropriate changes:

Line 208: Nutrient transport and accumulation in Malus hupehensis plants under different treatment conditions.

Line 553 (it was Line 515 before): The mitigation effects of exogenous dopamine on low nitrogen stress in Malus hupehensis.

Line 556: Beneficial effects of exogenous melatonin and dopamine on low nitrate stress in Malus hupehensis.

Line 560: Melatonin and dopamine mediate the regulation of nitrogen uptake and metabolism at low ammonium levels in Malus hupehensis.

Line 565: Exogenous melatonin improved potassium content in Malus under different stress conditions.

Line 571: Dopamine alleviates nutrient deficiency-induced stress in Malus hupehensis.

Line 642: Phosphate availability regulates root system architecture in Arabidopsis.

Line 644: Arabidopsis roots and shoots show distinct temporal adaptation patterns toward nitrogen starvation.

Line 659: Ionomic and metabolomic analyses reveal the resistance response mechanism to saline-alkali stress in Malus halliana seedlings.

Line 701: Exogenous myo-inositol alleviates salinity-induced stress in Malus hupehensis Rehd.

Line 707: Transcriptomic and metabolomic profiling provide novel insights into fruit development and flesh coloration in Prunus mira Koehne, a special wild peach species.

Reviewer 3 Report

Article written in clear and legible English with an introduction full of references and details. Excellent, detailed, numerous and innovative materials and methods to demonstrate the benefits of the experimental approach. Examples are the figures and tables in illustrating the results, which appear to be fully described with an abundance of parameters taken into consideration. Results consistent with what is known, reported in the literature. Conclusions in accordance with what they declare in the experimental design. Practical utility of the study in countering the effects of abiotic stress in conditions of nutritional deficiency with simple, valid, effective and economical methods in productive crops such as apple trees.

Int. J. Mol. Sci. 2018, 19, x; doi: FOR PEER REVIEW or Int. J. Mol. Sci. 2020, 19, x FOR PEER REVIEW, writing error

Figures 1a, 2a and 3a, dst error instead of mst

Tables S4, S6, S7 and S8 the comparison seems to be between CK and ST, ST and MST therefore up and down regolated the former and not the latter as stated in the text (line 262-267)

Line 409 LN, specify meaning

Line 475 VIP, specify meaning

Author Response

Responses to Reviewer #3

Thank you very much for the opportunity that you allowing us to revise our manuscript. We are encouraged by the very positive comments and also grateful to your valuable and constructive suggestions which would further improve the quality of our manuscript. We read the comments thoroughly and have modified the manuscript according to your comments. Changes and amendments are marked up using the ‘Track Changes’ in the revised manuscript. Point by point responses to comments are listed below.

Comment 1: Int. J. Mol. Sci. 2018, 19, x; doi: FOR PEER REVIEW or Int. J. Mol. Sci. 2020, 19, x FOR PEER REVIEW, writing error

Answer 1: Thanks for your comment. We have rewritten it as ‘Int. J. Mol. Sci. 2020, 19, x FOR PEER REVIEW’.

Comment 2: Figures 1a, 2a and 3a, dst error instead of mst

Answer 2: Thanks for your comment. We have fixed these errors in Figures 1a, 2a and 3a, and then reviewed the full text. All “DST” have changed to “MST”.

Comment 3: Tables S4, S6, S7 and S8 the comparison seems to be between CK and ST, ST and MST therefore up and down regolated the former and not the latter as stated in the text (line 262-267)

Answer 3: Thanks for your comment. The presentation of the two groups of comparisons in this manuscript is not uniformly annotated. We have checked the full text and carefully revised it. We have uniformly modified to: ST/CK, MST/CK, MST/ST.

Comment 4: Line 409 LN, specify meaning

Answer 4: Thanks for your comment. Due to our carelessness, we would like to express our sincerest apologies for the trouble caused by your understanding. The “LN” in this sentence should be changed to “ln” (ln = loge), and the formula refers to Liang et al (2018).

References

Liang, B.W.; Gao, T.T.; Zhao, Q., Ma, C.Q.; Chen, Q., Wei, Z.W.; Li, C.Y.; Li, C.; Ma, F.W. Effects of exogenous dopamine on the uptake, transport, and resorption of apple ionome under moderate drought. Front. Plant Sci. 2018, 9, 755.

Comment 5: Line 475 VIP, specify meaning

Answer 5:Thanks for your comment. VIP is an acronym for Variable Importance in Projection.